# Comparison of Efficacy in Patients with Metastatic Melanoma Treated with Ipilimumab and Nivolumab Who Did or Did Not Discontinue Treatment Due to Immune-Related Adverse Events: A Real-World Data Study

**DOI:** 10.3390/cancers13215550

**Published:** 2021-11-05

**Authors:** Morten Fink, Anders Schwartz Vittrup, Lars Bastholt, Inge Marie Svane, Marco Donia, Adam A. Luczak, Christina H. Ruhlmann, Louise Mahncke Guldbrandt, Ulrich Heide Koehler, Mette Lerche Winther, Eva Ellebaek, Charlotte Aaquist Haslund, Henrik Schmidt

**Affiliations:** 1Department of Oncology, Aarhus University Hospital, 8200 Aarhus, Denmark; anders.vittrup@oncology.au.dk (A.S.V.); louise.guldbrandt@midt.rm.dk (L.M.G.); uhk@braintrust-consult.dk (U.H.K.); henrschm@rm.dk (H.S.); 2Department of Oncology, Odense University Hospital, 5000 Odense, Denmark; Lars.Bastholt@rsyd.dk (L.B.); mette.winther@hotmail.dk (M.L.W.); 3National Center for Cancer Immuno Therapy, CCIT-DK, Department of Oncology, Copenhagen University Hospital, 2730 Herlev, Denmark; Inge.Marie.Svane@regionh.dk (I.M.S.); marco.donia@regionh.dk (M.D.); eva.ellebaek.steensgaard@regionh.dk (E.E.); 4Department of Oncology, Aalborg University Hospital, 9100 Aalborg, Denmark; adal@rn.dk (A.A.L.); cah@rn.dk (C.A.H.); 5Department of Clinical Research, University of Southern Denmark, 5230 Odense, Denmark; christina.ruhlmann@rsyd.dk

**Keywords:** ipilimumab, nivolumab, melanoma, immune-related adverse events, DAMMED, immunotherapy

## Abstract

**Simple Summary:**

This retrospective study of real-world patients with metastatic melanoma shows that discontinuing treatment with combination immunotherapy due to adverse events does not result in a poorer outcome compared to patients that did not discontinue due to toxicity. This is important knowledge for clinicians and patients, as discontinuing treatment may cause great anxiety for patients because they believe that it may limit the response.

**Abstract:**

Immune-related adverse events (irAEs) are very prevalent when treating patients with ipilimumab and nivolumab in combination, and 30–40% of patients discontinue the treatment for this reason. It is of high clinical relevance to investigate the consequences of discontinuing the treatment early since combination therapy with ipilimumab and nivolumab is the first line of treatment for many patients with metastatic melanoma. In this follow-up study, with real-world data from the nationwide DAMMED database, we investigated whether there was a difference in progression-free survival (PFS) and overall survival (OS) for patients who discontinued or did not discontinue treatment within the first four doses of treatment due to irAEs. In total, 448 patients were treated with ipilimumab and nivolumab. Of these, 133 patients discontinued due to irAEs in the induction phase. Using the Cox proportional hazards model, there was no significant difference in PFS when comparing the group that discontinued with the group that did not discontinue. The group that discontinued had a significantly longer OS than the group that received the full length of treatment. Therefore, we conclude that there is no significant negative impact on efficacy for patients who discontinue due to irAEs in the induction phase of combination immunotherapy for metastatic melanoma.

## 1. Introduction

Worldwide, the incidence of melanoma has been rising rapidly over the past few decades, and is now considered one of the most frequent solid cancers in fair-skinned populations [1]. Thus, in Denmark, it ranks as the fifth most common type of cancer [2]. An estimated 55,500 people worldwide die annually from melanoma [1,2,3,4].

Over the past decade, treatment of melanoma has evolved greatly due to an increased understanding of cancer biology and targeted therapy. Treatment of melanoma with antibodies directed against the immune checkpoint inhibitors CTLA-4 and PD-1 has shown a great increase in overall survival (OS), progression-free survival (PFS), and objective response rate (ORR), compared to other treatment modalities such as chemotherapy [5,6,7,8,9,10,11,12,13]. Additionally, immune checkpoint inhibitors administered as a combination of ipilimumab and nivolumab compared to ipilimumab as monotherapy have shown a further increase in OS, PFS, and ORR [5,6,8,9,14,15,16,17,18].

Grade 3–4 irAEs are often seen when treating patients with combination immunotherapy. Approximately 56% of patients treated with combination immunotherapy experience grade 3–4 adverse events. For patients treated with nivolumab monotherapy the proportion is 21%. Though the numbers are high, there seems to be no significant difference in quality-of-life assessment between the group that discontinues combination immunotherapy due to irAEs and those who do not [9,19,20,21,22].

Approximately 30–40% of patients receiving combination immunotherapy discontinue the treatment due to irAEs. It may cause anxiety for the patient to discontinue the treatment because they might fear that the disease will progress when they are not actively treated [9,19,20,23,24,25].

Hence, due to the high incidence of irAEs leading to patients discontinuing treatment with combination immunotherapy in the induction phase, it is of uttermost importance to the clinicians to investigate whether early discontinuation has an impact on OS, PFS, and ORR, compared to not discontinuing the treatment. Schadendorf et al. conducted a retrospective pooled analysis of the Checkpoint 067 and 069 protocols, which showed no significant difference in OS, PFS, or ORR for the two groups [20]. In this study, we seek to verify these findings with real-world data.

## 2. Materials and Methods

### 2.1. Patients and Study Design

This retrospective follow-up study was conducted to assess whether discontinuing treatment with ipilimumab in combination with nivolumab in the induction phase due to irAEs had an impact on OS, PFS, and ORR, compared to not discontinuing the treatment in the induction phase.

We extracted data from the Danish Metastatic Melanoma Database (DAMMED), a Danish nationwide database registering all patients with melanoma receiving medical oncological therapy in Denmark [26].

Baseline characteristics were logged at the first consultation with a clinician at the departments of oncology, and updated continuously regarding treatment and response, as well as progression and survival data.

All patients in this cohort had unresectable stage III or IV melanoma, and they received treatment between July 2016 and April 2021. Patients with cutaneous, mucosal, ocular, and melanoma with unknown origin are all included in the dataset. Also, the data includes all sites of metastases, including brain metastases.

Upon enrolment in the database, all patients submitted written consent to be included and can, at any time and for any reason, withdraw their consent. The study was accepted in the Regional Listing of Scientific Research with the file number 1-16-02-462-19.

### 2.2. Treatment Regimen

The induction phase consisted of 4 doses of a combination of 3 mg/kg of ipilimumab and 1 mg/kg of nivolumab intravenously, with each dose given 3 weeks apart.

The maintenance phase consisted of 6 mg/kg (max 480 mg) of nivolumab intravenously every 4 weeks for up to 24 months. Prior to April 2018, nivolumab was given as 3 mg/kg 2 weeks apart. Patients in this study received at least 1 dose of ipilimumab and nivolumab in combination.

The patients who discontinued due to irAEs were labeled as having discontinued if they failed to receive all 4 doses of combination immunotherapy in the induction phase. If these patients later resumed treatment with nivolumab as monotherapy, they were still considered as having discontinued. If a patient postponed a dose in the induction phase due to irAEs and resumed the induction phase later than 8 weeks after the prior dose, they were labeled as having discontinued treatment.

Patients who failed to complete the induction therapy due to progression of disease or death were included in the group of patients that did not discontinue treatment.

Non-discontinuing patients were defined as patients who received all 4 doses of induction therapy, without postponing a dose longer than 8 weeks, unless progression of disease was detected in the induction phase (Figure 1).

### 2.3. Assessments

The patients were assessed with either CT scans or FDG-PET/CT scans. A baseline CT scan or FDG-PET/CT scan was performed, followed by FDG-PET/CT or CT scans every 12 weeks for 2 years, and then every 6 months for an additional 3 years. A baseline CT or MRI scan of the brain was performed and if brain metastases were present, MRI scans were also performed with intervals as described above.

Besides the CT, FDG-PET/CT, and MRI evaluations, patients would receive clinical evaluations if necessary due to irAEs or clinical suspicion of progression. Blood samples were taken at baseline before every treatment, and repeated if there was suspicion of irAEs. In case of irAEs, a clinician would assess and treat the patient according to the guidelines made by the Danish Society for Clinical Oncology [27].

### 2.4. Statistical Analyses

PFS and OS were estimated using Kaplan–Meier statistics. Follow-up for the PFS analysis was obtained from the first treatment until progression of the disease. OS follow-up was obtained until the patient died.

In the survival analyses, we assess the variables: LDH level, defined as higher/lower than the upper limit of normal (ULN), CRP level, defined as higher/lower than 2 times the ULN, M1a versus M1b-d disease, M1d versus M1a-c disease, and cutaneous melanoma versus other diagnoses, to rule out potential confounders.

Differences between the 2 groups were assessed with a Cox proportional hazards model. Proportionality was validated by fitted survival curves. Univariate Cox proportional hazards models were made at first for all the variables logged in the database, to investigate factors that might confound the results. Factors that demonstrated significant differences in the univariate analyses would be further assessed with a multivariate Cox regression and a stratified Cox regression.

The ORR was assessed as the proportion of complete and partial responses in the 2 groups. Logging of data in the database was updated on 15 June 2021.

## 3. Results

### 3.1. Patient Characteristics

In total, 451 patients were enlisted in DAMMED as having received treatment with ipilimumab and nivolumab in combination. Three patients were excluded: one did not receive any dose of combination immunotherapy, and two were registered incorrectly. 133 (29.7%) patients did not receive all four doses of induction therapy due to irAEs and were categorized as patients who discontinued due to irAEs in the induction phase. This group included those who, after a treatment pause of more than 8 weeks, resumed therapy. The group of patients who did not discontinue treatment due to irAEs in the induction phase counted 315 (70.3%) patients. These patients were segregated into subgroups describing their status (Figure 1).

Patient characteristics were generally equally distributed between the groups, with a median age of 60.9 years for the group that discontinued therapy and 58.7 years for the group that did not.

There was a greater proportion in the non-discontinuing group that had stage M1d melanoma and CRP > 2x the ULN compared with the discontinuing group, suggesting that the non-discontinuing group might have had a greater proportion of patients with a poorer prognosis at baseline (Table 1).

Median follow up time was 14.4 months, with a range of 0 to 57.4 months.

### 3.2. Objective Response Rate (ORR)

ORR was assessed for the two groups. The discontinuing group had a proportion of investigator-assessed ORR of 54.9%, whereas the non-discontinuing group had a proportion of 39.0% of ORR (Table 2).

### 3.3. Progression-Free Survival

Median time to progression was 8.4 months (95%CI 6.0;12.8) and 7.7 months (95%CI 5.2;9.8) for the group that discontinued and that did not discontinue, respectively. At 24 months of follow-up, 32.8% were progression-free in the group that discontinued compared to 33.0% in the group that did not discontinue (Figure 2) (Table 3).

Univariate Cox analyses were made regarding LDH level higher versus lower than ULN, CRP level higher versus lower than 2x ULN, cutaneous melanoma versus other diagnoses, M1a versus M1b–d, and M1d versus M1a–c disease. In the univariate analyses, CRP level, M1a disease, and cutaneous melanoma diagnosis potentially confounded the estimate. Testing further with a multivariate Cox analysis, only CRP level and M1a were potential confounders, but when stratifying for the parameters, there was no contribution to the estimate of difference in PFS between the group that discontinued and the group that did not (Table 3).

### 3.4. Overall Survival

In the univariate analysis, there was a significant difference in OS between the group that discontinued and the group that did not discontinue (*p* = 0.02), with a median time not reached for the group that discontinued and 21.9 months (95%CI 16.68; 37.06) for those not discontinuing therapy (Figure 3).

Univariate Cox proportional hazards models were applied on potential confounding covariates (Table 4). These showed that having a high LDH level, CRP level, cutaneous melanoma, M1a, or M1d status potentially confounded the result.

Multivariate Cox analyses showed that LDH level and M1d status might be confounders to the result but stratifying for the factors did not change the estimate of OS between the group that discontinued and the group that did not discontinue treatment due to irAEs.

## 4. Discussion

Discontinuing treatment due to irAEs causes many patients emotional stress due to the general belief that discontinuing treatment early could limit the efficacy. Schadendorf et al. investigated this issue and found no difference between the group of patients that discontinued early due to irAEs and those who followed the treatment protocol longer than the induction phase [20]. This study aligns with these findings, showing that there is no difference between those who discontinued early due to irAEs and those who did not, regarding PFS. In contrast to prior research, this study shows that there is a significantly longer OS for the group that discontinued treatment due to irAEs. ORR was also higher in the group that discontinued treatment.

We included the patients who progressed or died during the induction phase in the group that did not discontinue due to irAEs, as was the case in a previous study [20]. This may have diminished the estimate of PFS and OS for the group that did not discontinue. 15% (*n* = 48) of the patients in the group that did not discontinue progressed in the induction phase, and 5% (*n* = 15) died. These patients are hard to interpret, since they probably diminish the PFS and OS for the group of patients that did not discontinue. On the other hand, they did not report any irAEs and did not qualify to be in the group that discontinued due to irAEs. None of the patients died from irAEs. The issue of early progressive disease and early death is an issue that must be taken into account when analyzing the data and should be analyzed further.

Contrary to the clinical study by Schadendorf et al. [20], there was no significant difference in the distribution of M1a–M1d patients between the groups. The two groups were homogenous regarding m-status, suggesting that m-status did not confound the result. In the study by Schadendorf, there was a greater proportion of patients with high LDH in the non-discontinuing group. Our study shows that LDH did not confound the result regarding PFS and OS, thus providing evidence that the prognosis for a patient that discontinues due to irAEs in the induction phase is not poorer than for a person that does not discontinue.

As a comment to the study conducted by Schadendorf et al. [20], it is argued by Horiguchi et al. [28] that using Kaplan–Meier statistics is an insufficient statistical method for the study. They argue that using a restricted mean survival time (RMST) results in a significantly increased OS for the group that discontinued treatment in the induction phase, indicating that serious irAEs and, therefore, discontinuation of therapy might be an actual marker of response to treatment [28,29]. Along with longer follow-up, RMST can also be used on the data from DAMMED.

The real-world setting of our analyses adds new insight to this clinical challenge. The group of patients from DAMMED are more inhomogeneous than the group of patients included in the clinical study by Schadendorf et al. [20,26]. The DAMMED database includes patients with ocular and mucosal melanoma, CNS metastases, and performance status >1. This adds real value to the results in a clinical setting. On the contrary, the group of real-world patients provides limitations to this study that should be taken into consideration when interpreting the results. The follow-up period is short, and longer follow-up time would be required to validate the findings. Additionally, due to the study being based on real-world data, there is possible residual confounding in the study design. Especially the patients that discontinue early due to irAEs and resume treatment with nivolumab monotherapy soon after are hard to interpret, thus, it is not evident whether the discontinuation marks an immune activation or the longer treatment with nivolumab causes a better outcome [10,30]. Investigation of subsequent PD-1 monotherapy should be done to further assess the benefit of resuming with monotherapy.

## 5. Conclusions

This study aligns with prior research and concludes that there is no significant negative impact on survival for metastatic melanoma patients who discontinue combination immunotherapy in the induction phase due to irAEs [20].

## Figures and Tables

**Figure 1 cancers-13-05550-f001:**
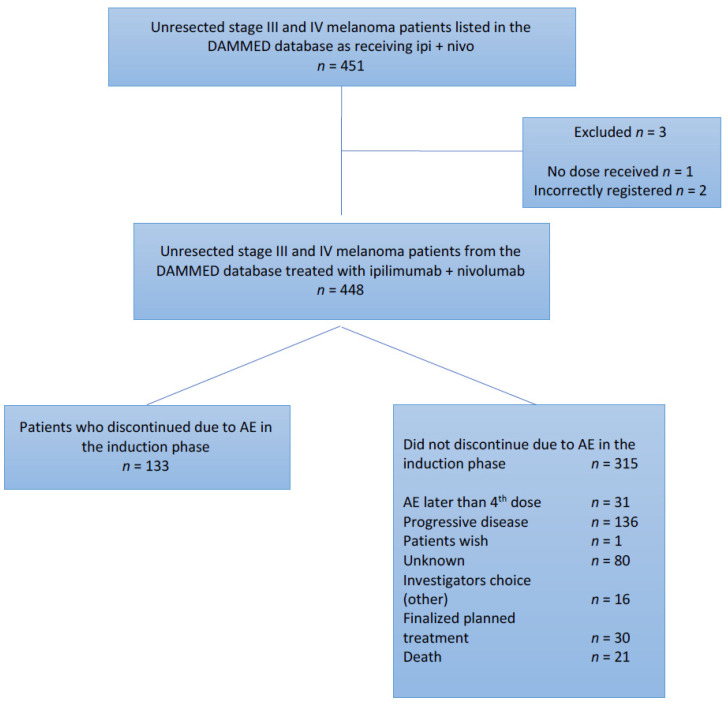
Patients divided into groups matching their discontinuation status and reason.

**Figure 2 cancers-13-05550-f002:**
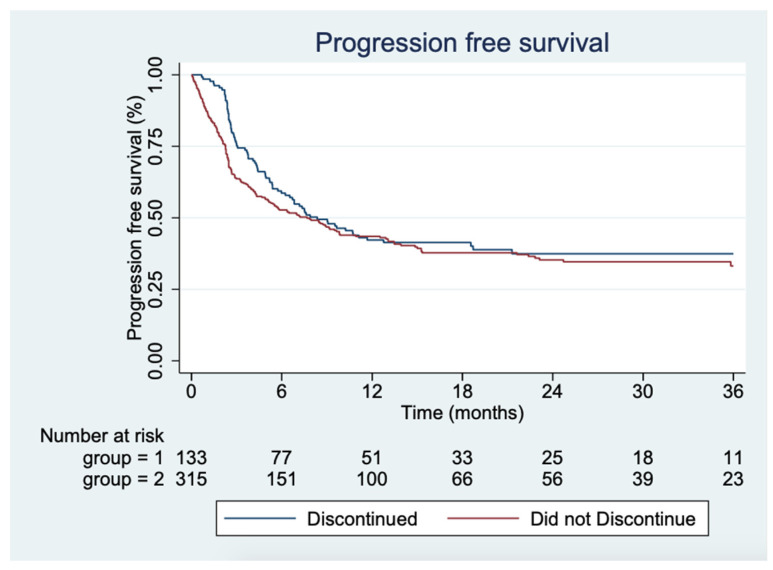
Kaplan–Meier plot showing the progression-free survival for the two groups, respectively.

**Figure 3 cancers-13-05550-f003:**
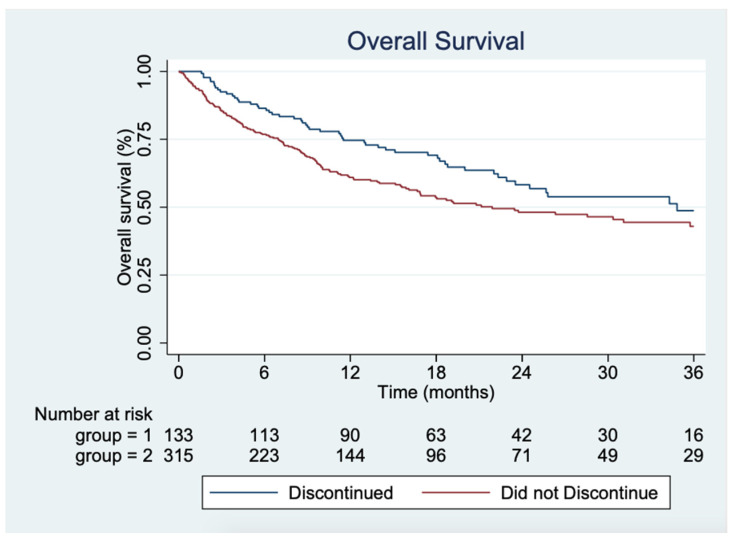
Kaplan–Meier plot showing the overall survival for the two groups, respectively.

**Table 1 cancers-13-05550-t001:** Baseline characteristics for the patients in the cohort.

Baseline Characteristics No. (%)	Patients Who Discontinued Due to irAEs in the Induction Phase *n* = 133	Patients Who Did Not Discontinue Due to irAEs in the Induction Phase *n* = 315
Gender		
-Male	69 (51.9%)	178 (56.5%)
-Female	64 (48.1%)	137 (43.5%)
Age		
->65	77 (57.9%)	196 (62.2%)
-<65	56 (42.0%)	119 (38.0%)
M stage		
-M1a	17 (12.8%)	36 (11.4%)
-M1b	15 (11.3%)	24 (8.0%)
-M1c	63 (47.4%)	152 (48.3%)
-M1d	38 (28.5%)	103 (32.3%)
LDH		
-≤ULN	58 (43.6%)	148 (47.0%)
->ULN	66 (49.6%)	148 (47.0%)
-Unknown	9 (7.0%)	19 (6.0%)
CRP		
-<2x ULN	106 (79.7%)	212 (67.3%)
->2x ULN	16 (12.0%)	69 (21.9%)
-Unknown	11 (8.3%)	34 (10.8%)
Melanoma diagnosis		
-Cutaneous	84 (63.2%)	206 (65.4%)
-Mucosal	7 (5.3%)	12 (3.8%)
-Ocular	22 (16.5%)	48 (15.2%)
-Unknown primary	20 (15.0%)	49 (15.6%)
BRAF mutation		
-Wild type	72 (54.1%)	149 (47.3%)
-Unknown	3 (2.3%)	21 (6.7%)
-V600E	51 (38.3%)	101 (32.1%)
-V600K	3 (2.3%)	13 (4.1%)
-V600x	4 (3.0%)	31 (9.8%)

**Table 2 cancers-13-05550-t002:** Response to treatment.

Variation	Patients Who Discontinued Due to Adverse Events *n* = 133	Patients Who Did Not Discontinue Due to Adverse Events *n* = 315
Objective response rate No. (%)	73 (54.9%)	123 (39.0%)
Best overall response		
No. (%)		
CR	33 (24.8%)	41 (13.0%)
PR	40 (30.1%)	82 (26.0%)
SD	21 (15.8%)	34 (10.8%)
PD	34 (25.6%)	141 (44.8%)
Unknown *	5 (3.7)	17 (5.4%)
Median time to progression (95%CI)	8.4	7.7
	6.0; 12.8	5.2; 9.8
Median time to death **	NR	21.9 16.7; 37.1

* TE (too early for assessment), unknown; ** from date of first treatment.

**Table 3 cancers-13-05550-t003:** Univariate and multivariate Cox regression analyses of potential confounding factors for PFS.

Variation		Univariate Analyses			Multivariate Analyses		
Covariates	No. of Patients	HR	95%CI	*p*	HR	95%CI	*p*
Group							
Non-discontinuing vs. discontinuing	315/133	1.19	0.92:1.55	0.19	1.22	0.93:1.60	0.15
LDH level							
high vs. low	213/207	1.20	0.94:1.53	0.15	-	-	-
CRP level							
>2× upper level vs. low	85/318	1.29	1.02:1.62	0.03	1.17	0.99:1.36	0.05
Melanoma diagnosis							
Cutaneous vs. other	290/158	0.82	0.72:0.92	0.00	1.06	0.89:1.25	0.51
M-status							
M1a vs. M1b–d	53/395	0.50	0.32:0.76	0.00	0.61	0.38:0.97	0.04
M1d vs. M1a–c	141/307	1.09	0.84:1.41	0.51	1.14	0.85:1.53	0.38
Cox models for group with stratifications							
Strata-m1a		1.23	0.93:1.61	0.14			
Strata-cutaneous melanoma		1.19	0.91:1.57	0.20			
Strata-CRP level high		1.18	0.89:1.56	0.24			

**Table 4 cancers-13-05550-t004:** Univariate and multivariate Cox regression analyses of potential confounding factors for OS.

Covariates	No. of Patients	Univariate Analyses			Multivariate Analyses		
		HR	95%CI	*p*	HR	95%CI	*p*
Group							
Non-discontinuing vs. discontinuing	315/133	1.45	1.06:1.99	0.02	1.48	1.07:2.07	0.02
LDH level							
high vs. low	213/207	1.29	0.96:1.72	0.08	1.35	1.01:1.80	0.04
CRP level							
>2× upper level vs. low	85/318	1.60	1.14:2.25	0.006	1.13	0.94:1.35	0.18
Melanoma diagnosis							
Cutaneous vs. other	290/158	0.72	0.54:0.96	0.03	0.99	0.68:1.46	0.98
M-status							
M1a vs. M1b-d	53/395	0.48	0.29:0.81	0.01	0.64	0.36:1.11	0.11
M1d vs. M1a-c	141/307	1.50	1.12:2.01	0.01	1.55	1.11:2.15	0.01
		Cox models for group with stratifications					
Strata–LDH level		1.49	1.08:2.08	0.02			
Strata–m1d		1.46	1.05:2.03	0.02			

## Data Availability

Restrictions apply to the availiability of these data. Data was obtained from DAMMED and are available with the permission of DAMMED.

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
