# Peer review of "Comparison of Efficacy in Patients with Metastatic Melanoma Treated with Ipilimumab and Nivolumab Who Did or Did Not Discontinue Treatment Due to Immune-Related Adverse Events: A Real-World Data Study"

_cancers, 2021, doi:10.3390/cancers13215550_

Round 1

Reviewer 1 Report

The study is worth the effort due to its real world setting. The groups need to be defined more precisely as suggested in the notes on text. I would suggest to enlarge discussion about the limitations of the study not only saying that "the results should be taken with caution". Please check references carefully and cite them appropriately

Author Response

Defining of groups has been made more clear and precise.

Errors regarding figure 1 has been corrected.

Definition and assessment of irAEs has been elaborated on, and has a reference to the guideline used by clinicians.

The discussion on limitations of the study, especially regarding early progressive disease and early death, has been extended further. 

There was a problem with the reference software. This has been corrected, and the number of references has been extended. Especially in the introduction.

Results regarding PFS has been rewritten and made more clear.

Reviewer 2 Report

In this paper, the authors presented real world data on melanoma patients receiving combination immunotherapy. They investigated whether early discontinuation of combination immunotherapy due to immune related adverse event had an impact on survival of patients compared to not discontinuing the treatment. It was a retrospective follow-up study. The results are clinically useful. The article is well-written.

1.Combination of anti-CTLA-4 ipilimumab and anti-PD-1 nivolumab has been increasingly used in recent years to treat patients with metastatic melanoma. The issue of whether the clinical outcome is better or worse in patients who have experienced an immune related side effect is of great interest to clinicians. In this retrospective follow-up study, the authors assessed whether discontinuing treatment with combination immunotherapy due to immune related adverse events had an impact on response rate, progression free survival and overall survival. The authors concluded that discontinuing treatment early due to immune related side effect did not limit the efficacy.

2.The topic is not original. A retrospective pooled analysis of clinical trial data on the same topic has already been published. However, it is clinically relevant that real-world data were presented in this article.

3.This adds new insight to the subject area.

4.The study was properly designed. The data were extracted from the Danish Metastatic Melanoma Database registering all patients with melanoma receiving medical oncological therapy in Denmark. Data of 448 patients treated with ipilimumab + nivolumab could be included in survival analysis. Description of the methodology was sufficiently detailed.

5.The interpretation of the results was correct; the conclusion was clear. The limitations of the study were identified and reported.

6.Relevant references are provided.

7.The Figures and Tables are clear and informative.

Author Response

Thank you for your comments and suggestions. We have made a spell check on the manuscript and corrected the mistakes.